# Antarctic Ice Sheet and emission scenario controls on 21st-century extreme sea-level changes

Thomas Frederikse [1,2]*, Maya K. Buchanan [3], Erwin Lambert [2], Robert E. Kopp [4], Michael Oppenheimer[5], D.J. Rasmussen[6] & Roderik S.W.van de Wal [2,7]

Uncertainties in Representative Concentration Pathway (RCP) scenarios and Antarctic Ice Sheet (AIS) melt propagate into uncertainties in projected mean sea-level (MSL) changes and extreme sea-level (ESL) events. Here we quantify the impact of RCP scenarios and AIS contributions on 21st-century ESL changes at tide-gauge sites across the globe using extreme-value statistics. We find that even under RCP2.6, almost half of the sites could be exposed annually to a present-day 100-year ESL event by 2050. Most tropical sites face large increases in ESL events earlier and for scenarios with smaller MSL changes than extratropical sites. Strong emission reductions lower the probability of large ESL changes but due to AIS uncertainties, cannot fully eliminate the probability that large increases in frequencies of ESL events will occur. Under RCP8.5 and rapid AIS mass loss, many tropical sites, including low-lying islands face a MSL rise by 2100 that exceeds the present-day 100-year event level.

[1] Jet Propulsion Laboratory, California Institute of Technology, 4800 Oak Grove Drive, California, CA 91101, USA. [2] Institute for Marine and Atmospheric Research Utrecht (IMAU), Utrecht University, P.O. Box 80011, 3508TA Utrecht, The Netherlands. [3] Climate Central, One Palmer Square, Suite 402, Princeton 08542 NJ, USA. [4] Department of Earth and Planetary Sciences and Institute of Earth, Ocean, and Atmospheric Sciences, Rutgers University, 71 Dudley Road, New Brunswick, NJ 08901, USA. [5] Department of Geosciences and the Woodrow Wilson School of Public & International Affairs, Princeton University, Robertson Hall, Princeton, NJ 08544-1013, USA. [6] Woodrow Wilson School of Public & International Affairs, Princeton University, Robertson Hall, Princeton, NJ 08544-1013, USA. [7] Department of Physical Geography, Utrecht University, P.O. Box 80115, 3508TC Utrecht, The Netherlands. *email: thomas.frederikse@jpl.nasa.gov

Future mean sea-level (MSL) changes will impact the height and frequency of extreme sea-level (ESL) events—the combination of tides and storm surges which could cause floods[1,2]. Any uncertainty in future MSL changes therefore propagates into uncertain ESL projections, which is a major concern for coastal communities, which rely on ESL projections to plan adequate and cost-effective adaptation measures[3,4]. In particular, low-lying tropical islands, which often lie only a few meters above sea level, are identified as being sensitive to moderate changes in MSL[5,6].

The major factors that cause uncertainty in future MSL changes are the emission scenario and the contribution from the Antarctic Ice Sheet (AIS)[7-9]. While in the Intergovernmental Panel on Climate Change (IPCC)'s Fifth Assessment Report (AR5)[7], the likely range for the AIS contribution under Representative Concentration Pathway (RCP) 8.5 was estimated to be between −6 to 12 cm for 2081–2100, relative to 1986–2005, recent ice sheet modeling and expert elicitation studies project AIS contributions on the order of multiple decimeters for this century, with high-end projections of AIS contributions exceeding 1 m in 2100[10-13]. Given the large spread in these projections, and our limited understanding of the underlying physics, the AIS projections are still characterized by a large degree of uncertainty, which hinders the quantification of the full probability density function of its contribution to MSL[9]. This uncertainty in MSL changes that cannot be fully quantified will lead to an uncertainty in the projected changes in ESL events that is not fully quantifiable either. MSL changes do not translate one-to-one into changes in the height and frequencies of ESL events[14], and therefore, the impact of the uncertain MSL changes will result in site-dependent ESL uncertainties.

In this paper, we compute projections of future ESL changes over a range of emission scenarios and possible AIS contributions to determine the impact of this uncertainty on changes in the height and frequency of ESL events. With these projections, we determine how MSL changes for these emission scenarios and the AIS contribution translate into local ESL changes. To do so, we apply extreme-value statistics and regional sea-level projections to a global set of tide-gauge sites[15]. We estimate ESL changes around mid-century (2046–2065) and end-century (2081–2100) for three RCP scenarios (RCP2.6, 4.5, and 8.5) and the possible range of AIS contributions based on a recent expert elicitation[13]. Because of the wide range of uncertainty, we do not assign a prescribed probability distribution to the AIS projection, but independently consider multiple scenarios. Therefore, we do not compute a most likely scenario or a single confidence interval for future ESL changes that only depends on the emission scenario. This approach deviates from other recent studies on MSL and ESL changes who assume specific probability distributions for the AIS contribution[8,9,16]. With this approach, we extend previous assessments[2,16-18] to fully account for the potential range of future ESL events at a global set of coastal sites. Recent developments in risk-assessment methods have highlighted the need to assess the full range of possible MSL and ESL scenarios in order to make robust decisions[19-21]. Given the strong possibility that multiple potential AIS distributions may be plausible for quite some time[22], such an account provides critical information for risk assessment purposes that consider the full range of possible ESL changes.

Reducing greenhouse gas emissions results in smaller changes in the occurrence and height of ESL events, but some sites, particularly in the tropics will see a large increase in the number of ESL events, even halfway the twenty-first century. Under high-end emission scenarios and AIS contributions, almost all sites see large changes in ESL changes, while for some tropical sites, mean sea level could reach or surpass the height of the present-day 100-year event at the end of the twenty-first century.

## Results

**Emission scenarios and the AIS contribution.** We use the IPCC's AR5 MSL projections[7] for all components except for the AIS. For the AIS, we assume a possible range of the AIS contribution between 0 and 45 cm in 2046–2065, and a possible range between 0 and 150 cm in 2081–2100, based on the expert elicitation from the ref. [13]. All projections are relative to 1986–2005. For clarity, we combine the RCP2.6 and RCP8.5 scenarios with the low-emission and high-emission projections from ref. [13]. For each of these scenarios, we select the AIS contribution that corresponds to the upper and lower bound of the likely range (within the 17–83% confidence interval), and round the contribution to the nearest 5 cm. All results are available for RCP4.5 and for the full range of AIS contributions from the Supplementary Information.

For 2046–2065, the spread in projected MSL changes for each emission scenario and accompanying likely range of AIS contributions is limited to a few decimeters (Fig. 1), which is caused both by the limited spread between the emission scenarios and because the spread in the aforementioned likely range of the AIS contribution[13] is still small. Towards the end of the twenty-

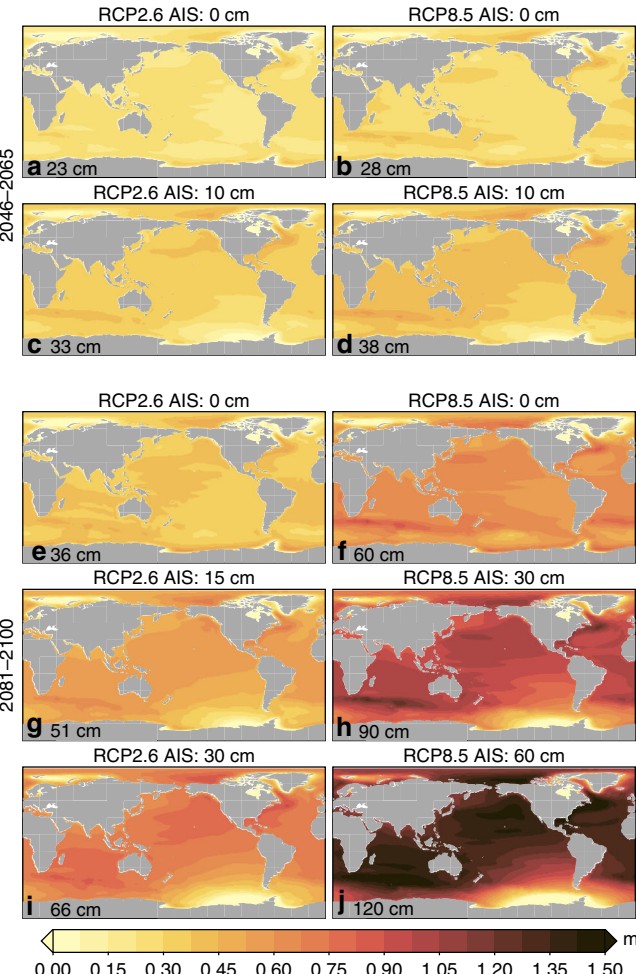

**Fig. 1 Mean sea-level change scenarios.** Selection of median regional IPCC AR5 MSL scenarios relative to the 1986–2005 baseline combined with various AIS contributions. The title of each panel shows the RCP scenario and the AIS contribution in cm. **a–d** 2046–2064. **e–j** 2081–2100 for varying RCP scenarios and AIS contributions. The number in the bottom left corner shows the corresponding global-mean sea-level change. See Supplementary Figs. 1–3 for all considered emission scenarios and accompanying uncertainty estimates.

first century, for many regions, the spread between the MSL scenarios in Fig. 1 is on the order of a meter. The differences between the RCP2.6 and RCP8.5 emission scenarios are of a similar magnitude as the differences between the low-end and high-end AIS contribution within each emission scenario: both halfway and towards the end of the century, the global-mean sea-level changes under RCP2.6 with a high end AIS contribution are comparable to the changes under RCP8.5 with the low-end AIS contribution. The regional MSL patterns vary between these scenarios: for the high-end AIS contributions, the spatial patterns caused by gravitational, rotational, and deformation (GRD[23]) effects due to AIS mass loss become one of the dominant spatial features, which results in above-average MSL changes in the North Atlantic, North Pacific and Indian Oceans (Fig. 1).

**Characteristics of extreme sea-level events**. The height and return frequency of ESL events can be expressed as a return curve, which links a specific height above mean sea level to the expected number of exceedance events (Fig. 2d). We estimate the return curve by fitting the generalized pareto distribution (GPD) to observed extreme events[1,24], as described in the Methods section. This method is equivalent to the method used in the ref. [25]. The GPD is described by three parameters: the location parameter, the height which roughly corresponds to a regularly-occurring extreme event, the scale parameter, which describes the variability of the exceedances, and the shape parameter, which controls the curvature and determines the behavior at very long return periods[26]. See the methods section for a full description of the GPD and its parameters.

The properties of the return curve differ from site to site. Figure 2 shows return curves for two tropical islands and extratropical coastal cities. The Galveston and Saipan records contain ESL events driven by tropical cyclones, which are visible as peak ESL events. The upward curvature of the return curve results in unbounded return levels for the lowest frequencies, and corresponds to a positive shape parameter. Due to the fact that

over the record period only a few tropical cyclones have been observed, the return period of these events is poorly constrained, which results in a poorly-constrained return curve and large confidence intervals. Brest and Gan have not been exposed to tropical cyclones over the length of the tide-gauge record and are characterized by bounded return curves that are well-constrained by observational data. These stations have a negative shape parameter and the shape and scale parameter have a small uncertainty. For Brest, the height difference between typical ESL events and MSL is largely determined by tides, which translates into a large location parameter. Gan only sees a small tidal range and has a nearly flat return curve, which corresponds to a small scale parameter.

Figure 3 shows that the GPD parameters and ESL characteristics vary from place to place. Some regions show coherent ESL characteristics: a notable feature is the effect of tropical cyclones on the coast of the Gulf of Mexico and the Southeastern United States, but also some tropical islands. For places that are subject to tropical cyclones, the ESL characteristics are similar to Galveston and Saipan. Regions with large tidal amplitudes, such at the Western European and Northern Australian coastline and Alaska are distinguishable by large location parameters, and behave similar to Brest. Places that are often prone to storm surges, such as Alaska and the North Sea coast generally see a large height difference between relatively common and rare events, which corresponds to a high scale parameter. The combination of these parameters again results in a spatially heterogeneous map of present-day 100-year return heights (Fig. 3e). The European, North Australian, and Northwest American coasts show return heights that often exceed four meters, while many islands in the Pacific and Indian Ocean have a 100-year return height on the order of 1 m.

The uncertainty in the derived return heights (Fig. 3f) depends on both the scale and shape parameters and the record length, and is generally on the order of one to three decimeters, except for places with a positive shape parameter, where the uncertainty is about an order of magnitude larger. The spatial variations in

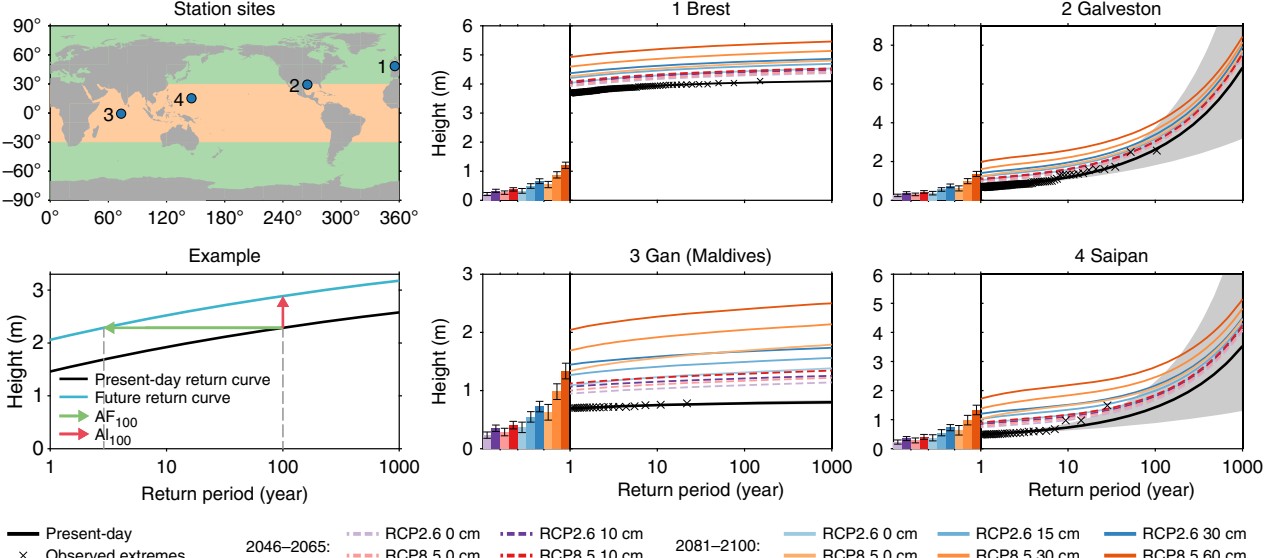

**Fig. 2 Example return curves. a** The station sites and the regions that correspond with the tropics (orange) and extratropics (green). **d** An example return curve and the definition of the 100-year allowances (AI$_{100}$) and amplification factors (AF$_{100}$). The other panels show the present-day return curves and the curves after future MSL changes relative to 1986–2005 mean. The MSL changes are shown in the bar chart next to each panel. The numbers in the bar chart show the location (loc), scale (scl), and shape (shp) parameters of the return curve for the corresponding station. The shaded area shows the [5−95]% confidence interval of the present-day return curve, which is computed from the uncertainties in the estimated extreme-value distribution parameters, see methods for more details. The y-axis differs between sub-panels. Note that for Brest and Gan, the confidence interval is displayed, but small, compared to the scale of the graph.

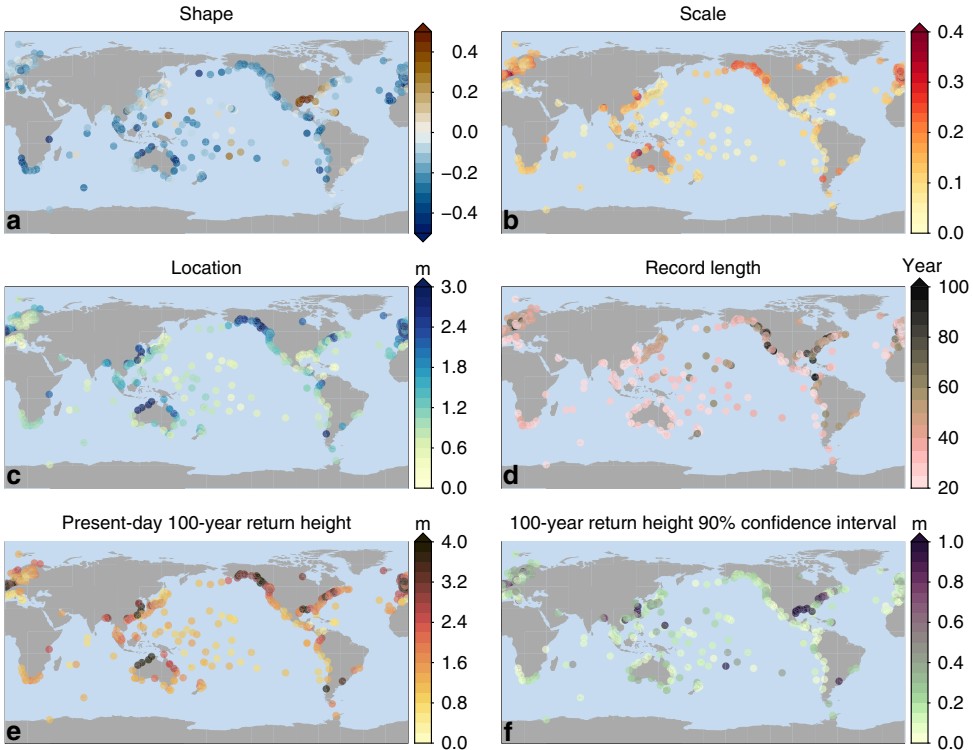

**Fig. 3 Properties of the tide-gauge sites.** The shape (**a**), scale (**b**), and location (**c**) parameters of the GPD distribution, the record length, expressed in the number of unique years in each record (**d**), the present-day 100-year return height (**e**), and its uncertainty (**f**), expressed as the width of the 5–95% confidence interval. See Supplementary Fig. 5 for the confidence intervals for the shape and scale parameters.

the present-day return heights (Fig. 3e) are generally larger than the uncertainties at individual sites, except for regions with positive shape parameters, and these spatial variations are not an artifact of the uncertainties when fitting an extreme-value distribution.

**The impact of mean sea-level changes on extreme sea-level events.** Changes in regional MSL will modify the frequencies and heights of ESL events. Under the stationarity assumption (i.e., future ESL characteristics relative to MSL do not change over time), a MSL change results in a vertical shift of the return curve (Fig. 2b). The shift of the return curve changes the frequency of ESL events, from which we derive the frequency amplification factor (AF)[17,27], which is the amplification of the expected return frequency of a present-day ESL event. The height change of ESL events with a fixed return frequency after a change in mean sea level is the sea-level rise allowance (Al)[28,29].

Both future MSL changes and present-day return curves come with an uncertainty. A symmetric uncertainty in the MSL change (the vertical axis in Fig. 2) translates into an uncertainty in the return period for a specific height (the horizontal axis in Fig. 2) that is skewed towards shorter return periods[29]. The uncertainty in the present-day return curve further enhances this skewness. To account for this uncertainty, we present future return curves as the mean, or best estimate, of this skewed distribution. These mean return periods are shorter than the median return periods relating to the median MSL change. Both uncertainties therefore prescribe that both AF and Al are larger than expected from the median MSL change. For sites where the uncertainty in future MSL changes is large and where small MSL changes result in large amplification factors, this asymmetry causes a larger difference between the allowance and the expected MSL change. The allowance can be used as a guideline for required changes in the

design height of coastal infrastructure under an uncertain change of mean sea level.

In the remainder of this paper, we will discuss the future changes of ESLs in terms of the AF of the present-day 100-year event ($AF_{100}$) and the allowance of the 100-year event ($Al_{100}$). The impact of MSL change on the allowances and amplification factors depends on a multitude of factors: next to the magnitude of the MSL change, the ESL characteristics and their uncertainties result in a station-specific change in the allowance and the amplification factor. Figure 2 illustrates this dependence: for Gan and Brest, as well as for other sites with small scale and shape parameters, a small MSL change causes a large $AF_{100}$, while the same scenarios will induce a smaller $AF_{100}$ for both Galveston and Saipan. This dependence also allows to qualitatively estimate the impact of MSL changes for places for which no tide-gauge observations are available, but knowledge about typical ESL behavior is available. For places where surges from tropical cyclones or other storms cause large and rare ESL events compared to more common events, MSL changes result in smaller AFs than for places with a flatter return curve.

**Global changes in amplification factors.** The combination of the spatial variability in expected MSL changes and extreme-value characteristics leads to spatially-varying AFs for each scenario (Fig. 4). As discussed before, smaller scale and shape parameters cause larger AFs for a given MSL change, and the spatial variations in AFs deviate from the spatial pattern in the driving MSL changes.

Halfway through this century, under the scenario with the smallest MSL rise (RCP2.6 and no AIS contribution), 25% of the tide-gauge sites are projected to face a $AF_{100}$ of 100 or larger, which means that the present-day 100-year event is projected to occur at least annually on average. In the tropics (between 23.43°S

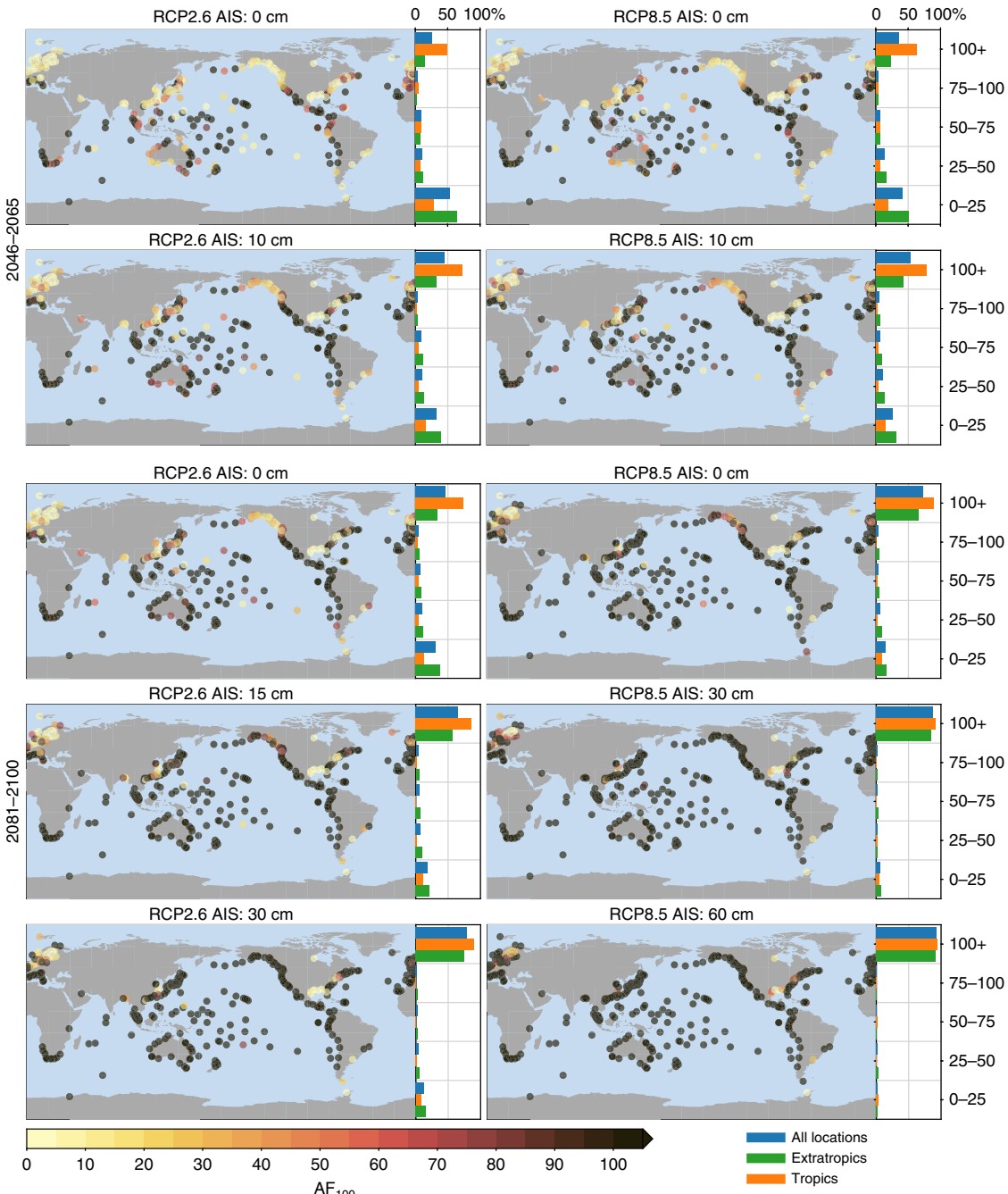

**Fig. 4 Frequency amplification factors for the 100-year extreme sea-level event.** The bars on the right show the histogram of the AFs, separated into stations in the tropics (latitude between ±23.44°), extratropics (latitude outside ±23.44°), and all stations. See Supplementary Figs. 6 and 7 for the frequency amplification factors for all scenarios and AIS contributions.

and 23.43°N), 50% of the sites have a $AF_{100}$ of 100 or larger, which is not only caused by the above-average MSL changes in the tropics, but also because many of these stations have a relatively flat return curve. For the RCP8.5 scenario and no AIS contribution, these numbers increase modestly: 35% and 63% of the global and tropical sites, respectively will see a $AF_{100}$ of 100 or larger. Adding a 10 cm AIS contribution, which corresponds to the upper end of the likely range for this scenario, results in a further increase in frequency amplification factors: under RCP2.6, almost half (44%) of all sites, and the majority (72%) of the tropical sites will face an $AF_{100}$ of 100 or higher. Under the RCP8.5 scenario with a 10 cm AIS contribution, this is the case for more than half (52%) of all sites, and for more than three-

quarters (77%) of the tropical sites. The number of stations with an expected $AF_{100}$ of 100 or larger is higher for RCP2.6 with 10 cm AIS contribution than for RCP8.5 without any AIS contribution, which means that, akin to the MSL scenarios, the AIS uncertainty is on the same level as the spread between scenarios halfway through the century. While many tropical sites will face large AFs halfway through the century, even for the most optimistic scenario, the AFs for large parts of the European and North American coastlines remain relatively modest, even for higher emission scenarios and AIS contributions. However, it must be noted that AIS contributions of multiple decimeters beyond the likely range as depicted in Fig. 1 cannot be ruled out, which could lead to high AFs for these coastlines as well.

Towards the end of the century the differences between the scenarios become larger: under the no-AIS RCP2.6 scenario, the number of sites with a $AF_{100}$ of 100 or higher rises to 46% globally (73% for the tropics), while for RCP8.5, most sites (72% globally, 89% for the tropics) will face a $AF_{100}$ of 100 or more. These numbers indicate that, even without any AIS contribution, many sites will face large increases in the frequency of ESL events at the end of the century, while for the tropics, even under the no-AIS RCP2.6 scenario, 50% of the sites sees such large increases already halfway through this century. For the many extra-tropical sites, notably the coastlines of North America, Europe, and parts of Asia, the expected AF does depend on the emission scenario: in the most optimistic case, 34% of the sites faces an $AF_{100}$ of 100 or

higher, while in the RCP8.5 scenario, or the RCP2.6 scenario with an AIS contribution of 15 cm, more than half of the stations sees such an increase. Under the higher-end AIS contribution, also the vast majority of the extratropical sites faces an $AF_{100}$ of 100 or more. Again, RCP2.6 combined with the high-end AIS contribution result in larger AFs than RCP8.5 with a low-end AIS contribution.

**Global changes in allowances.** The spatial pattern of the allowances (Fig. 5) shows less spatial heterogeneity than the AF pattern, and follows to a large extent the spatial pattern of the MSL scenarios (Fig. 1), although the allowance exceeds the MSL

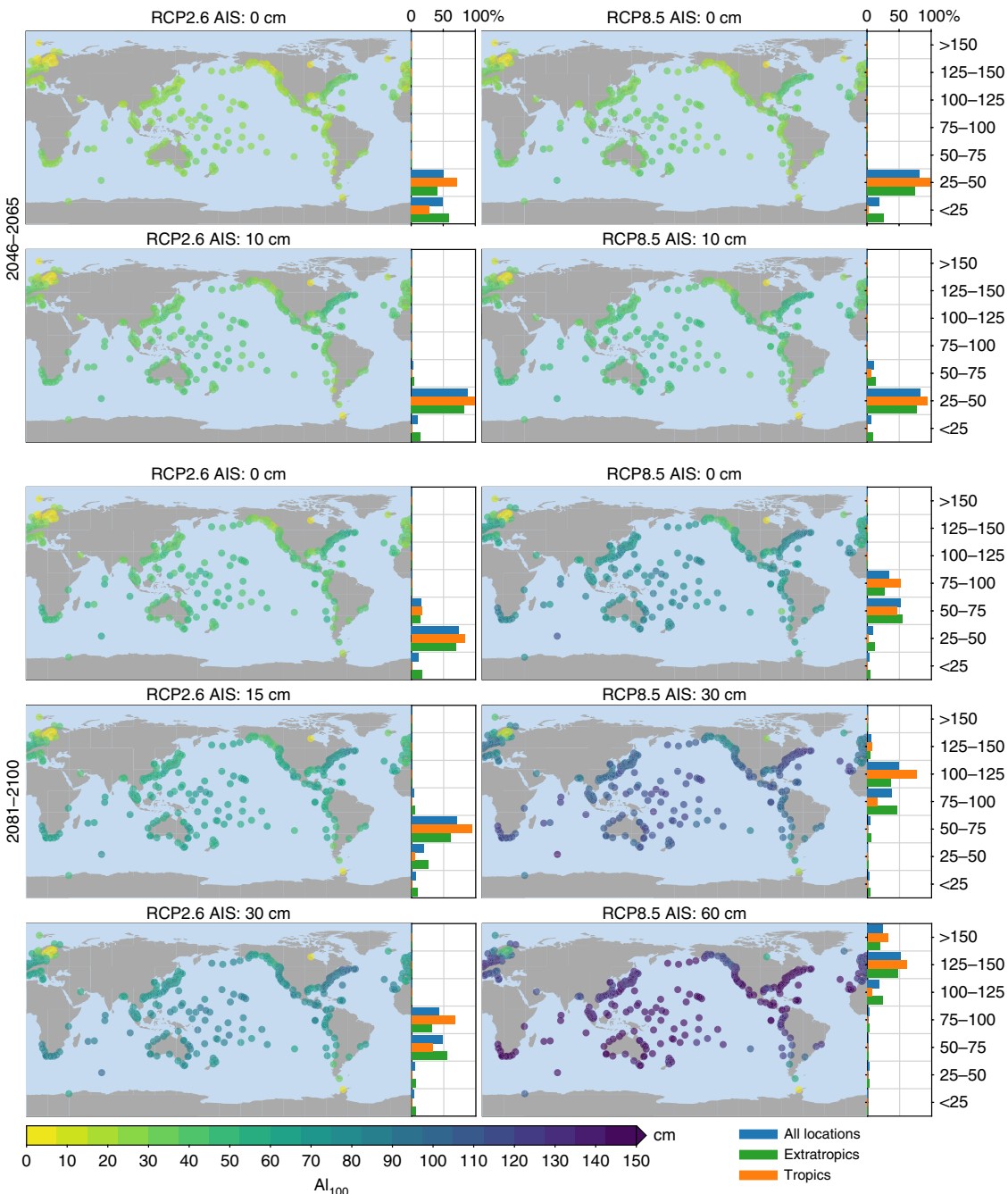

**Fig. 5 Allowances for the 100-year extreme sea-level event.** The bars on the right show the histogram of the allowances, separated into stations in the tropics (latitude between ±23.44°), extratropics (latitude outside ±23.44°), and all stations. See Supplementary Figs. 8 and 9 for the allowances for all scenarios and AIS contributions.

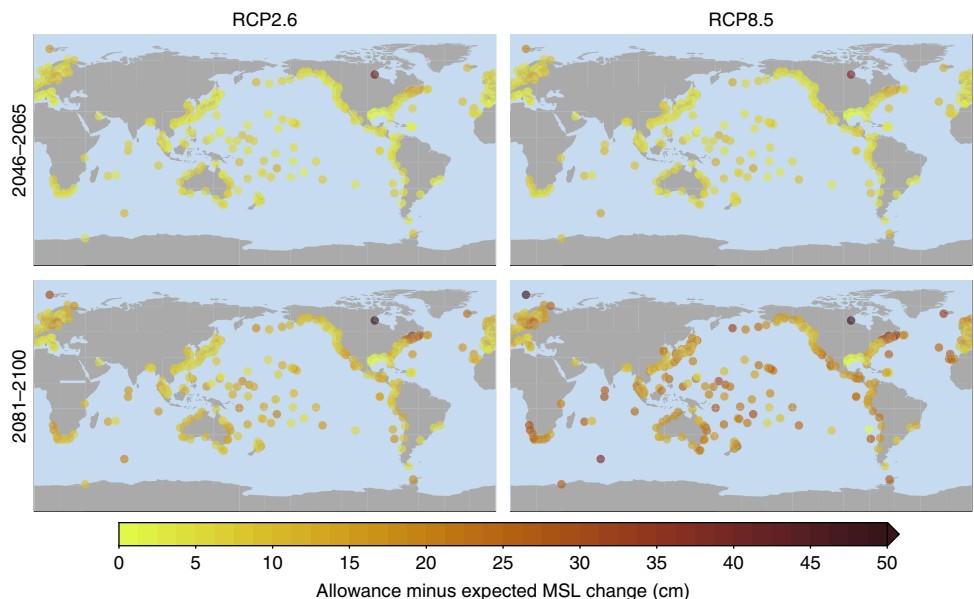

**Fig. 6 The 100-year allowance minus the expected mean sea-level change.** The allowance minus MSL change only depends on the RCP scenario, since we do not assign an uncertainty to the AIS contribution.

change due to the uncertainty in the GPD parameters and in the MSL scenario (Fig. 6)[29]. Despite the different origin of the spatial patterns, akin to the AF case, the tropics generally see higher allowances than the extratropics, and the spread between the scenarios halfway through the century is limited compared to the end of the twenty-first century. At the end of the century, the allowances also depend strongly on the AIS contribution, as indicated by the $Al_{100}$ for different end-century scenarios (Fig. 5). Under RCP2.6 and no AIS contribution, the median $Al_{100}$ is 39 cm. Again, the tropics have a higher median $Al_{100}$ of 44 cm. For RCP8.5 and no AIS contribution, the median $Al_{100}$ rises to 70 cm (global) and 76 cm (tropics). Adding an AIS contribution of 30 cm to the RCP8.5 scenario results in a median $Al_{100}$ of 1.02 (global) and 1.10 m (tropics), and for a 60 cm AIS contribution, the median $Al_{100}$ becomes 1.35 and 1.46, with many sites, both in the tropics and along the Eastern US coast, requiring an allowance of 1.5 m or more. The spatial variations in the allowances are primarily caused by the the regional MSL patterns, since the differences between the $Al_{100}$ and the expected MSL changes are on the order of a decimeter (Fig. 6, and the spatial pattern of these differences are smaller than the spatial variations in the MSL projections. Therefore, in contrast to the AF case, these allowances can be extrapolated to sites without a tide gauge with a comparably small margin of uncertainty, which runs from a few centimeters in for RCP2.6 halfway through the century to a maximum of a few decimeters for RCP8.5 towards the end of the century. Note that this difference is independent of the AIS contribution in our case, since there we do not attribute an explicit uncertainty to the AIS contribution.

**Mean sea level versus extreme events.** For many sites, including many low-lying islands, the present-day 100-year return height is on the order of 1 m above mean sea level (Fig. 3). This situation implies that for higher-end scenarios, the expected MSL change is of the same magnitude as the height of a 100-year event under present-day conditions. Under these scenarios, mean sea level could reach or surpass the present-day 100-year level. This is for example the case for Gan and Saipan, as depicted in Fig. 2. This implies that for these sites, areas that are at the present-day 100-year level and considered safe will be located in the inter-tidal

zone if no adaptation measures will be taken. We can express this situation by computing the return period that corresponds to the present-day return height equal to the expected MSL change. Figure 7 shows this ratio for the end of the century. This ratio strongly depends on the RCP scenario and the AIS contribution. Towards the end of the century, for the most optimistic scenario (RCP2.6 without any AIS contribution), only one percent of the sites will see a MSL change corresponding to the present-day level with a return period of 1 year or lower, while for RCP8.5 without any AIS contribution, this percentage rises to 5%. Adding an AIS contribution of 60 cm, this number rises to 49%, with 22% of the stations will face a MSL change equal to the present-day 100-year event or higher. For the tropics, this number is 41%, and includes many low-lying islands in the Pacific and Indian Ocean. These numbers imply that under these high-end scenarios, mean sea level at these sites will be higher than the highest recorded ESL event.

## Discussion
This analysis shows that a strong reduction in greenhouse gas emissions reduces the allowances and the amplification factors, but even under the RCP2.6 scenario, some sites, particularly in the tropics will see large amplification factors halfway this century. We find that independent of the chosen emission scenario and AIS contribution, more than a quarter of the sites will face an $AF_{100}$ of 100 or more already halfway through this century. Towards the end of the century, more than half of the sites will face such frequency amplification, except for the most optimistic emission scenario without any AIS contribution. For higher emission and AIS scenarios, large frequency amplification factors will occur not only at more places, but also earlier in time, and for a high-end emission scenario combined with a 10 cm AIS contribution, which is considered the upper end of the likely range mid-century[13], more than half of the stations already face an $AF_{100}$ of 100 or more.

Amplification factors can be used as a metric to quantify expected future changes in the occurrence of contemporary ESL events, but they do not provide information on the impact of MSL changes on the height of future ESL events. Complementing AFs with allowances provide a more rounded picture of impact

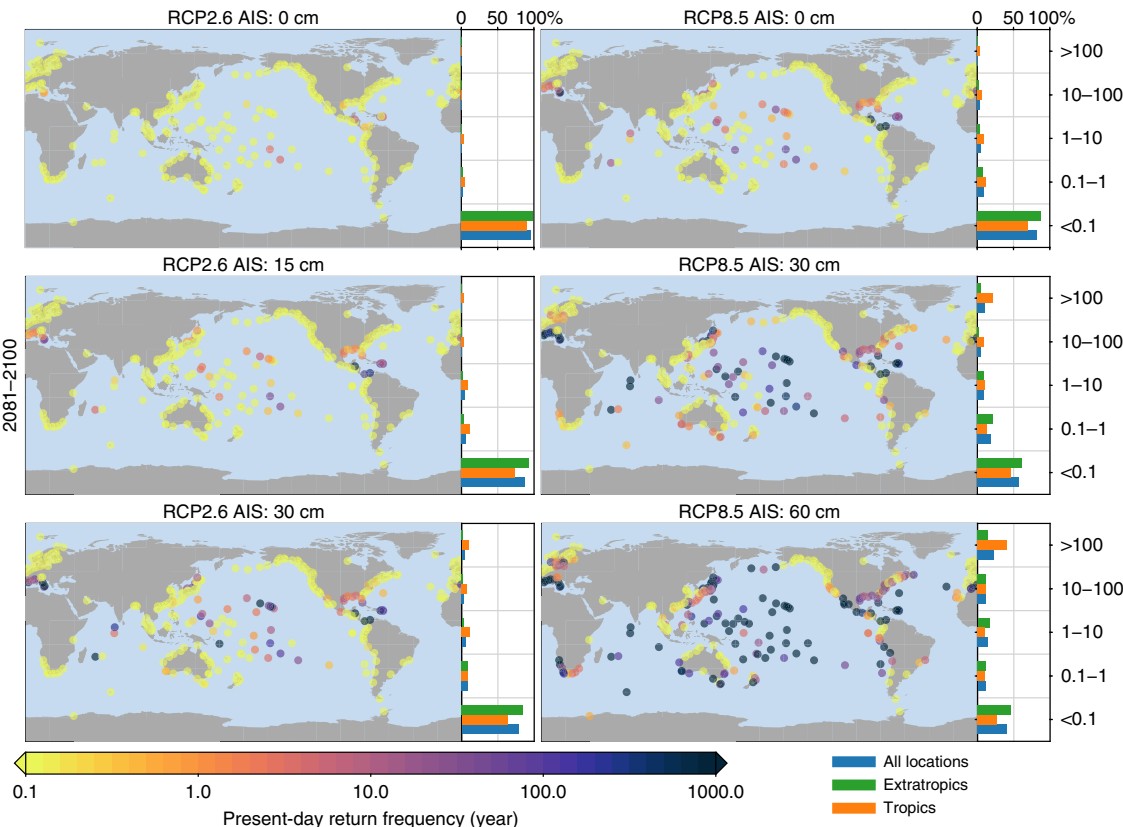

**Fig. 7 Projected mean sea-level changes versus present day extreme sea-level events.** Each panel shows the present-day return period that that has a return height equal to the expected mean sea level above present-day mean sea level in 2081–2100 for each scenario. A value of 10 years means that the expected MSL change is the same as the present-day return height with a return period of 10 years. The bars on the right of each plot show a histogram of the tropical, extratropical, and all stations.

and risk because AFs contain information on the expected future height of ESL events in addition to frequency. However, a complete picture of the flooding risk would also require information on exposure and vulnerability of people and infrastructure[30], but this is beyond the scope of the current study.

The allowances required to keep the exceedance risk at present-day levels do depend on the emission scenario and AIS contribution, even halfway through the century. Towards the end of the century, depending on the scenario, typical allowances range between a few decimeters to values well above a meter. We also find that the ESL changes for RCP2.6 with a higher-end AIS contributions are similar to the ESL changes for RCP8.5 with a lower-end AIS contribution. This similarity implies that even with a substantial reduction of greenhouse gas emissions, large frequency amplification factors and allowances on the order of 50–100 cm cannot be ruled out.

The estimates of future changes in ESL events come with a number of caveats related to the usage of tide-gauge records and the statistical method. Since tide-gauge observations are typically low-pass filtered by stilling wells or other filtering methods, and subsequently sampled at lower frequencies, waves are removed from the observed water levels, and wave setup is poorly represented in the tide-gauge records, although it may contribute significantly to ESLs[31]; thus our ESLs are approximately extreme coastal still-water levels. In addition, the length of tide-gauge records is typically a few decades, and as such, events with low probabilities could be under-represented or over-represented, leading to a bias in the return curve[32]. For example, tropical cyclones are typically low-probability events, and records in cyclone-prone regions could be too short to contain a cyclone,

which could bias the results for these short records. In the application of the statistical method, a number of choices are made which may impact the results. We have chosen the GPD as extreme-value distribution over other commonly-used distributions based on a block-maxima approach, such as the Gumbel and generalized extreme-value distribution. The main reason for this choice is that the GPD uses all available extreme-value events and is not limited to the highest event per month or year. Furthermore, due to the presence of a the shape parameter allows more accurate model of low-frequency events, compared to the Gumbel distribution. A comparison between multiple distributions show good performance for the GPD[2]. The choice of a threshold to define the fitted GPD is a trade-off between bias and variance. As a consequence, the GPD fit may not adequately describe ESLs at all sites, even though the chosen value (the 99.7th percentile) has been shown to perform well in a global analysis[2]. Due to these choices, and because of the specific AIS contribution, the estimated future return curves and the resulting amplification factors and allowances could differ from previous assessments (e.g., ref.[27]). Another assumption in the statistical method is that of stationarity, i.e., the assumption that the local ESL climatology does not change in time. Locally, observations have reported temporal changes in ESL climatology[33,34]. In addition, hydro-dynamic models coupled to earth-system models suggest that future MSL changes may impact tides and surges in specific regions[35,36], hence modifying local ESL climatology. Changing wind patterns and changes in the strength and probability of tropical cyclones could also occur in a warming climate, and affect local ESL characteristics[37]. However, both observations[38,39] and model studies[18] indicate that changes in MSL are the major

contributor to past and future changes in ESLs on a global scale. Besides temporal changes in climatology, we also do not account for inter-annual variability in MSL, which modulates ESL events. As such, variability in present-day ESL events in different sites may be underestimated, which could lead to an overestimation of frequency amplification factors and allowances. In contrast, both frequency amplification factors and allowances may be underestimated since we do not explicitly include an uncertainty related to the AIS contribution, do not consider a full interdependency in the uncertainty of all contributors[40,41], and do not account for the possibility of a heavy-tailed uncertainty for the Greenland Ice Sheet or other individual contributing processes, which could cause larger frequency amplification factors and allowances than reported here. For some locations, AFs in previous work (e.g., Buchanan et al.[27]) differ from values presented here due to differences in the Antarctic contribution, GPD parameters, and methods for detrending and declustering.

Due to the spatial variations in both the expected local MSL changes and in the local ESL characteristics, the expected amplifications and allowances are dependent on the specific site. While the amplification factors show a strong dependence on the local ESL characteristics, the allowances largely follow the spatial variations in the MSL projections. Similar to the early emergence of climate-driven temperature changes[42,43], the tropics, which include many low-lying islands, are the first to see large amplification factors, and require above-average allowances. While the high amplification factors are mostly driven by the typical ESL climatology, the high allowances are to a large extent caused by the above-average projected MSL changes for this region. Since many of these islands only lie a few meters or less above present-day MSL, the required allowances under high-end AIS contributions and RCP scenarios are on the same order of magnitude as the height of these islands above present-day mean sea level. Towards the end of the century, for many sites with relatively small present-day return heights, including many tropical islands, the expected MSL change under high-end scenarios will exceed the present-day height with a return period of 100 years or less. This change implies that for these sites and scenarios, a sea level that is now considered an extreme event will become mean sea level.

Both future emissions and the fate of the AIS are poorly constrained, and as a result, allowances typically vary from a few decimeters to one-and-a-half meter by the end of the century within the likely range of AIS contributions, while changes that are substantially larger cannot be ruled out. A reduction of greenhouse gas emissions substantially lowers the required allowances, and limits the likely range of the AIS contribution, although the uncertain AIS contribution could still cause allowances on the order of 50–100 cm. However, a reduction will not only result in smaller allowances, but will also increase the available time for coastal areas to adapt to changing ESL return curves, although for some sites, especially many open-ocean islands, large increases in return frequencies will occur within a few decades, regardless of the Antarctic contribution to MSL and the emission scenario.

## Methods

**Mean sea-level scenarios**. The MSL scenarios depicted in Fig. 1 are based on the scenarios presented in the IPCC AR5[7], except for the Antarctic contribution. The AR5 projections provide estimates of the processes related to steric expansion and ocean dynamics (ocean), the surface mass balance (SMB) and the dynamic ice-sheet response (Dyn) for the Greenland Ice Sheet (GrIS), the contribution of Glacial Isostatic Adjustment (GIA), and changes in terrestrial water storage (TWS). Since AR5, several studies addressed the dynamic contribution of the Antarctic Ice Sheet (AIS) resulting from marine ice sheet instability[11,12] and its combination with hydro-fracturing of ice shelves and ice cliff instability[10]. These processes could lead to multiple meters of sea-level rise over the next few centuries. However, ice-sheet dynamics remain a source of debate, and therefore, we have chosen, instead of using a single probability distribution, to compute ESL changes for a range of possible AIS scenarios. We consider possible AIS contributions between 0–45 cm for 2046–2065, and 0–150 cm for 2081–2100, where the upper bound is chosen to match the 95th percentile from[13]. The figures in the main text show the mean and bounds of the likely range (17–83% confidence intervals). The projections for all scenarios are available from the Supporting Information.

The projections are expressed as changes in regional relative sea level, which are changes in sea surface height relative to the solid earth. The AR5 scenarios account for the regional MSL changes due to GIA and the GRD effects resulting from ice-level and TWS-related sea-level changes. To account for GRD effects for the AIS contribution, we computed the elastic sea-level fingerprint using the pseudo-spectral approach[44], with rotational feedback included[45]. We assume that all Antarctic mass loss occurs at Thwaites Glacier basin. This basin currently shows the largest mass-loss rate[46] and has the potential of ice-sheet instabilities[47,48]. Other nearby regions along the Amundsen Sea Sector are vulnerable as well, but since almost all tide-gauge sites considered in this study are in the far field, small changes in the geometry of the mass loss are unlikely to significantly change the results[49]. Supplementary Fig. 4 shows the normalized fingerprint used for the AIS contribution.

The AR5 projections come with an estimate of the uncertainties. In the original AR5 projections, the probability distribution for some contributions, especially the Greenland Ice Sheet SMB (GrIS SMB), were considered skewed. We assumed a normal distribution, with standard deviations computed from averaging the upper-bound and lower-bound standard deviation as listed in the AR5 report. We chose this approach, because no full distribution is available. The downside of this approach is that the possible heavy-tailed distribution is not explicitly accounted for, which could result in an underestimation of the allowances. Uncertainties for the individual contributors are assumed to be independent and added in quadrature to obtain the total uncertainty. The exception are the ocean and Greenland SMB contributions which are considered fully correlated, since both are heavily dependent on temperature changes, and hence the sum of the standard deviations of those quantities was added in quadrature to the total uncertainty. Since we use a range of possible AIS scenarios, we do not attach an uncertainty to each AIS scenario. The final expression for the total uncertainty reads:

$$\sigma_{total} = \sqrt{(\sigma_{ocean} + \sigma_{SMB\ GrIS})^2 + \sigma_{Glaciers}^2 + \sigma_{GIA}^2 + \sigma_{GWD}^2 + \sigma_{Dyn\ GrIS}^2} \quad (1)$$

The total uncertainties for all sea-level scenarios can be found in Supporting Information Fig. S3.

**Tide-gauge data**. The tide-gauge data used in this study originate from the GESLA2 dataset[15]. Data provided on frequencies higher than hourly are averaged to hourly values. For each station, we use station-years with a data availability of 70%. We limit the extreme-value analysis to stations with at least 20 years of data which are at least 70% complete. When a single site has multiple records in the GESLA2 database, the longest record has been chosen, which results in a total number of 460 stations. From each station-year the annual mean is subtracted to remove the effects of long-term sea-level trends and variability. Since some stations have duplicate entries in the GESLA2 database, for stations that are within a 3 km radius, we select the station with the longest record.

**Extreme-value analysis**. We define an extreme sea-level (ESL) event as an event for which sea level reaches a height above the annual mean MSL that exceeds the 99.7th percentile of hourly sea-level observations. The choice of the specific percentile is arbitrary, but the chosen value shows stable results in regional and global studies[2,24]. Since an event can last for multiple hourly observations, the data has to be de-clustered. We use a declustering time of three days between consecutive events to enforce this requirement. From these observations of sea-level extremes, the return frequency $N(i)$ of the $i$th-highest ESL can be estimated using the standard Weibull formula[24]:

$$N(i) = \frac{i}{1+n}, \quad (2)$$

with $n$ the total number of observation-years. To compute return heights of return frequencies that surpass the tide-gauge record length, extreme-value statistics are used. An extreme-value distribution has been fitted to the de-clustered sea-level extremes. Following the recommendations of ref. [2], we use the GPD. The GPD uses all extreme-value information in the record and the shape parameter allows for improved tail modeling, compared to the annual-maximum approach that is often employed when the Gumbel or generalized extreme-value (GEV) distributions are used. For the GPD, the probability density function $f(z)$ for height above MSL $z$ is

$$f(z|z>\mu) = \begin{cases} \frac{1}{\lambda}\left(1 + \xi\left(\frac{z-\mu}{\lambda}\right)\right)^{-\left(1+\frac{1}{\xi}\right)} & \text{for } \xi \neq 0 \\ \frac{1}{\lambda}e^{-\left(\frac{z-\mu}{\lambda}\right)} & \text{for } \xi = 0 \end{cases} \quad (3)$$

with location parameter $\mu$, scale parameter $\lambda$ and shape parameter $\xi$. The location parameter $\mu$ corresponds to the height of the 99.7th percentile of the hourly observations. $f(z)$ is only defined for $z > \mu$, and we can only determine return levels for heights larger than the location parameter. After de-clustering the extreme events, this threshold coincides to the height of an event that occurs on average six times a year, and thus can be viewed as a commonly-occurring extreme event. This

parameter is estimated as the height $z$ that corresponds to the 99.7th percentile. The scale parameter $\lambda$ and shape parameter $\xi$ are estimated using the method of maximum likelihood. The scale parameter $\lambda$ determines the slope of the return curve: a large $\lambda$ corresponds to a large slope. The shape parameter $\xi$ determines the curvature of the tail of the return curve. The return curves in Fig. 2b, e have a negative shape parameter, and as a result the curve has an upper bound for large return periods. A shape parameter of zero would result in a straight return curve, akin to the Gumbel distribution. A positive shape parameter corresponds to an upward curvature and an unbounded return height for very low return frequencies Fig. (2c, f). Supplementary Fig. 5 shows the estimated median and 90% confidence intervals for the shape and scale parameters.

The associated exceedance probability of height $z_0$, given $z_0 > \mu$ is

$$E(z > z_0 | z > \mu) = \begin{cases} \left(1 + \xi\left(\frac{z-\mu}{\lambda}\right)\right)^{-\frac{1}{\xi}} & \text{for } \xi \neq 0 \\ e^{-\left(\frac{z-\mu}{\lambda}\right)} & \text{for } \xi = 0 \end{cases}. \qquad (4)$$

Assuming that the events are Poisson distributed, the return frequency, which is the expected number of events per year, follows from multiplying $E(z > z_0 | z > \mu)$ with $\zeta$, the average number of events per year[26]:

$$N(z > z_0) = \begin{cases} \zeta\left(1 + \xi\left(\frac{z-\mu}{\lambda}\right)\right)^{-\frac{1}{\xi}} & \text{for } \xi \neq 0 \\ \zeta e^{-\left(\frac{z-\mu}{\lambda}\right)} & \text{for } \xi = 0 \end{cases}. \qquad (5)$$

Under a change in mean sea level, we assume that $\lambda$ and $\xi$ stay constant, but the location parameter $\mu$ is augmented with the sea-level change: $\mu_f = \mu_h + \Delta MSL$ with $\mu_f$ and $\mu_h$ the future and historical location parameter and $\Delta MSL$ the change in mean sea level. From Eq. (5), we compute return curves that link return frequencies to specific water levels above MSL. Both the GPD parameters and the MSL change come with an uncertainty, and the changes in the water levels and return frequencies of ESL events will be uncertain too. To incorporate this uncertainty into the projected changes in ESL events, we determine the return curve that shows for each return height the expected return frequency under the uncertain model and MSL parameters. The resulting projected return curve always generates higher return frequencies for a given height than the return curve based on median parameters. This difference is the largest for sites where the MSL uncertainty is large compared to the uncertainty in ESL characteristics and the scale parameter, which describes the ratio between the height of rare extreme events versus common extreme events. This approach returns a single return curve that represents expectations under the current uncertainties in all parameters. To compute this expected return curve, we use a Monte-Carlo approach, following the approach from the ref. [25,27,28]. For the GPD parameters, we obtain a variance-covariance matrix for the uncertainties in the parameters $\lambda$ and $\xi$ using the observed inverse Fisher information matrix. We generate $10^4$ random pairs of $\lambda$ and $\xi$ with mean and standard deviation based on the maximum likelihood fit and the accompanying variance–covariance matrix. We also generate $10^4$ Monte-Carlo samples of local MSL change from the mean and standard deviation of each scenario. For each ensemble member, we compute the return curve Eq. (5). From the resulting ensemble of return curves, we determine the mean (or expected) return frequency for each return water level, which gives the expected return curve. From this curve, the amplification factor and allowance can be directly determined. The combination of a minimum record length of 20 years and a threshold parameter of 99.7% guarantees that we can always compute a return curve for events with a return period of 1 year or more. We do not account for sub-annual events, and amplification factors that reach the sub-annual level will result in a "larger than annual" output.

## Code and data availability

The AR5 sea-level projections have been downloaded from http://icdc.cen.uni-hamburg. de/1/daten/ocean/ar5-slr.html. The region mask for Thwaites Glacier has been obtained from http://icesat4.gsfc.nasa.gov/cryo_data/ant_grn_drainage_systems.php. This mask has been produced by Zwally, H. Jay, Mario B. Giovinetto, Matthew A. Beckley, and Jack L. Saba in 2012 at the GSFC Cryospheric Sciences Laboratory. The GESLA2 high-frequency tide-gauge data set[15] has been downloaded from http://gesla.org/. All figures have been produced using the Generic Mapping Tools (GMT). The mean sea-level scenarios and the estimated amplification factors and allowances for each scenario are available from https://doi.org/10.5281/zenodo.3550108. The codes to compute the mean sea-level scenarios, perform the extreme-value analysis, and compute the resulting return curves, allowances, and amplification factors are available from https://github.com/thomasfrederikse/ECWL.

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

## Acknowledgements

Parts of this research (T.F.) were carried out at the Jet Propulsion Laboratory, California Institute of Technology, under a contract with the National Aeronautics and Space Administration. T.F. also acknowledges the NWO ALW NPP program. R.E.K. was supported in part by National Science Foundation grant ICER-1663807 and National Aeronautics and Space Administration grant 80NSSC17K0698. E.L. acknowledges funding from the INSeaPTION project, which is part of ERA4CS, an ERA-NET initiated by JPI Climate, and funded by FORMAS (SE), BMBF (DE), BMWFW (AT), IFD (DK), MINECO (ES), ANR (FR) with co-funding by the European Union (Grant 690462).

## Author contributions

T.F., E.L. and R.vdW. conceived the idea of the study, T.F. and E.L. ran the computations, and T.F., M.K.B., E.L., R.E.K., M.O., D.J.R. and R.vdW. wrote the paper.

## Competing interests

The authors declare no competing interests.
