## [Peer Review File · Nature Communications]

Reviewers' comments:

Reviewer #1 (Remarks to the Author):

The manuscript presents estimates for extreme coastal water level changes for the 21st century using the concept of sea level allowances. The authors present their estimates in very brief form and mainly state summary statistics. In this form the paper misses the chance to comprehensibly inform the reader on the consequences of IPCC-AR5 style sea-level rise for extreme coastal water change. In my view, the paper needs a detailed presentation of the regional results, a clearer motivation and critical discussion of the choice of sea level projections, and a clearer analysis how sea level projections shape the changes in extreme coastal water levels. The concept of sea level allowance is after all tailored to inform on the regional (or local) level and the authors should better explore this. The paper has also large overlaps with the published results of Slangen et al. 2018. A distinction to the work is not discussed though authorships partly overlap. This raises questions of originality of the work, which need to be addressed. See also Hunter et al. 2017. I also struggle with the choice of 4 main-text figures and 12 supplementary figures, with SI figures repeating partly results of the main text figures. This should be better balanced. My major points follow below:

- 1) The authors assume future sea level change following IPCC AR5 except for the Antarctic ice sheet. An ad-hoc estimate is used for the Antarctic contribution. While IPCC AR5 projections are an obvious first choice, the Antarctic contribution needs better justification. Just mentioning that it is deeply uncertainty is not enough to motivate this. While Deconto & Pollard 2016 is used to motivate deep uncertainty in the work, the DP16 results show a strong dependence of Antarctic ice loss on global warming while the authors assume complete independence. I also think that an overview figure of the global mean sea level projections is necessary and would help the reader to quickly grasp the authors' choices. This could be timeseries or a bar chart.
- 2) The paper largely retains a global focus presenting summary statistics while extreme coastal water levels (ECWL) are relevant on a local level. Though the abstract states – and the methodology is capable of – deriving ECWL estimates at tide gauge locations, the paper does not present these. This global focus also leads to weak conclusions in the abstract. Larger allowances in the tropics than elsewhere is already known and not new (e.g. Slangen et al. 2018, Vousdoukas et al. 2018). The large increase of ECWL even under modest sea level rise is also already known and a consequence of the shape of extreme value distributions. The value of the paper would increase through an in-depth presentation and discussion of the regional estimates, beyond the four locations that are presented in Fig. 2. The data for each tide gauge station (Fig. S6-S10) should be presented in a Supplementary Table or in other digital form to be of value for local stakeholders.
- 3) On the level of global summaries, I think it would be really valuable to better understand the relation between sea level change and ECWL change on an analytical level. The paper presents the numbers, but leaves the reader alone with guessing relations between slr and ECWL change. This could be exemplified, for example, with a figure showing both global sea level and aggregated ECWL evolution and divergence. Maybe even an analytical expression could be derived for local slr vs ECWL change. Vousdoukas et al. 2018 is a good source of inspiration for presenting results.
- 4) The authors use the generalized pareto distribution to fit extreme water level distributions instead of the Gumbel distribution used in most of the earlier works, e.g. Hunter et al. 2017. The sensitivity of the paper's results to this choice is not discussed. This makes it difficult to compare to earlier works. Please add. The authors also discuss that many places AF100 of 100 or larger, meaning that former 100y events occur sub-annually. How does the methodology deal with sub-annual events? How does it handle mean sea level rising above the 100y ECWL? Does it occur at some places?
- 5) I would favor the code underlying this work to be presented open source on one of the known hosting platforms (github ...) , with a clear description (readme) of all steps to replicate the analysis, including the sources of additional data downloads.

Minor comments:

L32: The maximum AIS contribution is stated as 75cm. This number is not used in any of the main text figures and text. This needs to be explained.

L47-49: I do not understand this sentence and the "free-board" without further explanation. Please expand on this concept. Maybe merge from Methods.

L58-59: why are there no confidence intervals shown for Brest and Gan? Reads like a physical explanation, but it is just a fit to a function, right?

Fig. 2: The lines for Brest and Gan are very flat. This means that for a certain (small) increase of mean sea level, a mapping to a (shorter) return period cannot be established. This may be just an effect of the y-scale of the figure or a weakness of the methodology. It should be discussed.

L74: Relatively? Do not understand. "slight" missing?

L76: How can I have a factor AF100 of 100 or more? Do you resolve sub-annually?

L80: This statement is already clear from the author's choice of future AIS contribution, which has the same magnitude as the sum of all other components.

L89-L90: Meridional overturning circulation: How does it enter into calculations?

L108-109: It would also reduce the risk of a high Antarctic contribution. This is not reflected in the author's choices of climate-independent Antarctic ice loss.

L129: Do you mean Fig. S4 here?

L148: Tide gauge data: how many locations are used finally?

L163-L164: Reference to figures seem wrong.

L186-L204: This discussion can easily be missed. It may be better placed at the end of the main text.

References:

DeConto, R. M., & Pollard, D. (2016). Contribution of Antarctica to past and future sea-level rise. *Nature*, 531(7596), 591.

Hunter, J. R., Woodworth, P. L., Wahl, T., & Nicholls, R. J. (2017). Using global tide gauge data to validate and improve the representation of extreme sea levels in flood impact studies. *Global and Planetary Change*, 156, 34-45.

Slangen, A., van de Wal, R. S., Reerink, T. J., de Winter, R. C., Hunter, J. R., Woodworth, P. L., & Edwards, T. (2017). The impact of uncertainties in ice sheet dynamics on sea-level allowances at tide gauge locations. *Journal of Marine Science and Engineering*, 5(2), 21.

Vousdoukas, M. I., Mentaschi, L., Voukouvalas, E., Verlaan, M., Jevrejeva, S., Jackson, L. P., & Feyen, L. (2018). Global probabilistic projections of extreme sea levels show intensification of coastal flood hazard. *Nature Communications*, 9(1), 2360.

Reviewer #3 (Remarks to the Author):

Review of Frederikse et al. "The role of emission scenarios and Antarctica in 21st century extreme water level changes".

This paper uses data from tide gauges to derive return levels for extreme water level events, then considers how these return levels are likely to change with sea-level rise according to certain RCP projections. Antarctic ice loss is treated separately, as scenarios for varying losses from one location in Antarctica.

Results are presented as the "frequency amplification factor" AF100, ie are saturated when a "1/100" year event becomes an annual event, or as AI100 "sea-level rise allowance" - change of 100yr extreme level. This presentation may be useful in communicating the global risk. However the paper contains little discussion of these results.

I recommend that for publication in Nature Communications the paper be revised to either highlight the mechanisms behind the spatial variation, and hence enable generalisation to data-poor regions, or else to provide a data set or maps more directly useful for coastal planners, perhaps exploring some interesting cases.

The explanations around figure 2 are very good - perhaps the most useful part of the paper. Since the shape of the return curve is dependent on tidal range, which is relatively well known (eg line 63), and on cyclone likelihood, a map of tidal range and cyclone tracks may be useful. Can fig 3 then be predicted from these? Or could you indicate which gauges globally are similar in characteristics to the examples 1/2/3/4? Maybe plot all gauges that are like (1) on one map, and all those like (4) on another? For example, is the block of similar results along Alaska/west Canada a direct result of the high tidal range?

There is a significant problem with the presentation of the data as a scatter-plot at tide gauges (Fig 3, 4, S5-S12), in that even zooming in you cannot see some data points because they are obscured by their neighbours. For example there are points on the US east coast, and in Europe, just peeking out, which are very different from their neighbours, and it is not clear which they are or whether they are important.

The scatterplot order appears to be alphabetical by gauge, so is pseudo-random with location, but this method of plotting can give a false impression of spatial patterns and is useless for looking up individual locations in crowded areas. Can the authors improve the plot to avoid this overlap? An online version of the plots with smaller dot size compared to the coastline would help, as would use of transparency.

The uncertainty of the return curves is very variable between gauges, because of the varying length of data as well as the different weather and tide patterns. This is to some extent captured in figure S6 but this is hard to interpret. Could you plot the uncertainty at the 100yr level, to give some indication in the confidence in these results by location? Are those locations that differ from their neighbours those with high uncertainty, or is there a lot of local spatial variability?

It is not very clear why the ice-loss scenarios are all based on changes to Thwaites glacier. I'm not a cryosphere expert - are these a fair reflection of the likely 21st century scenarios? Would other likely scenarios substantially change the result? What about Greenland? Is Greenland ice loss included in the other RCP maps?

Return levels calculated at tide gauges could include many local effects, including tidal range and wetting-and-drying of low-lying land. These are ignored in the calculations, which assume a linear addition of the extra water. This is probably necessary in a global study but should be explicitly stated. Likewise the assumption of no future change in extremes due to tropical cyclones should be made very explicit.

Can you avoid the unusual acronym "GRD"? SAL (self-attraction and loading) is more commonly used, although to be fair that doesn't encompass rotation.

typo fig 2 "Maldives".

Please find below a point-by-point response to the reviewer's comments. Our response is in blue, and orange text denotes changes in the manuscript. All references to line numbers refer to the line numbering in the version of the manuscript with marked changes.

The data supplement, which includes all mean sea-level and ESL scenarios and the underlying code can be found here: <https://thomasfrederikse.stackstorage.com/s/MsHhwz2nZjxUEGs>. These scripts and data will be put in an open repository when the paper has been accepted for publication. Please find below a detailed response to the comments of both reviewers.

Reviewer 1

The manuscript presents estimates for extreme coastal water level changes for the 21st century using the concept of sea level allowances. The authors present their estimates in very brief form and mainly state summary statistics. In this form the paper misses the chance to comprehensibly inform the reader on the consequences of IPCC-AR5 style sea-level rise for extreme coastal water change.

In my view, the paper needs a detailed presentation of the regional results, a clearer motivation and critical discussion of the choice of sea level projections, and a clearer analysis how sea level projections shape the changes in extreme coastal water levels. The concept of sea level allowance is after all tailored to inform on the regional (or local) level and the authors should better explore this. The paper has also large overlaps with the published results of Slangen et al. 2018. A distinction to the work is not discussed though authorships partly overlap. This raises questions of originality of the work, which need to be addressed.

See also Hunter et al. 2017. I also struggle with the choice of 4 main-text figures and 12 supplementary figures, with SI figures repeating partly results of the main text figures. This should be better balanced. My major points follow below:

1) The authors assume future sea level change following IPCC AR5 except for the Antarctic ice sheet. An ad-hoc estimate is used for the Antarctic contribution. While IPCC AR5 projections are an obvious first choice, the Antarctic contribution needs better justification. Just mentioning that it is deeply uncertainty is not enough to motivate this. While Deconto & Pollard 2016 is used to motivate deep uncertainty in the work, the DP16 results show a strong dependence of Antarctic ice loss on global warming while the authors assume complete independence. I also think that an overview figure of the global mean sea level projections is necessary and would help the reader to quickly grasp the authors' choices. This could be timeseries or a bar chart.

We agree with the reviewer that the contribution from the Antarctic Ice Sheet needs extra attention in the paper. Based on the results from a recent expert elicitation (Bamber et al. 2019), we have updated the range of Antarctic Ice Sheet contributions. This allows us to use a mean and likely range of AIS scenarios for each epoch and RCP scenario, as well as an upper bound of a possible AIS contribution. We have added a figure (Figure 1), which shows the regional MSL scenarios, as well as the corresponding GMSL changes.

Furthermore, the aim of this paper is to show how ESL events depend on both the emission scenario and the AIS contribution, and we do not assign a likelihood to each specific scenario or AIS contribution, except for that the results are summarized for the likely range from Bamber et al. (2019). Because of this approach, we do not have to consider the dependence between the AIS contribution and the emission scenario. The manuscript we presented here indeed has some overlap with the work from Slangen et al. (2018), although there are multiple key differences. The key difference is that in Slangen et al. (2018), as well as in multiple other studies on the impact of sea-level changes on extremes, it is assumed that the probability distribution of the contribution from the Antarctic Ice Sheet is known. The recent discussions on the onset of the instability of the Antarctic Ice sheet shows the presence of substantial deep uncertainty of this contribution, and we discuss the impact of this deep uncertainty on changes in extreme sea levels. We have re-written the introduction paragraph to explain the purpose of this manuscript and discuss the approach we are using.

In this paper, we compute projections of future ESL changes over a range of emission scenarios and possible AIS contributions to determine the impact of this uncertainty on changes in the height and frequency of ESL events. With these projections, we determine how MSL changes for these emission scenarios and the AIS contribution translate into local ESL changes. To do so, we apply extreme-value statistics and regional sea-level projections to a global set of tide-gauge locations. We estimate ESL changes around mid-century (2046-2065) and end-century (2081-2100) for three RCP scenarios (RCP2.6, 4.5, and 8.5) and the possible range of AIS contributions based on a recent expert elicitation. Because of this deep uncertainty, we do not assign a prescribed probability distribution to the AIS projection, but independently consider multiple scenarios. Therefore, we do not compute a most likely scenario or a single confidence interval for future ESL changes that only depends on the emission scenario. This approach deviates from other recent studies on MSL and ESL changes who assume a specific probability distribution for the AIS contribution.

With this approach, we extend previous assessments to fully account for the potential range of future ESL events at a global set of coastal locations. Recent developments in risk-assessment methods have highlighted the need to assess the full range of possible MSL and ESL scenarios in order to make robust decisions. Given the strong possibility that multiple potential AIS distributions may be plausible for quite some time, such an account provides critical information for risk assessment purposes that consider the full range of possible ESL changes.

2) The paper largely retains a global focus presenting summary statistics while extreme coastal water levels (ECWL) are relevant on a local level. Though the abstract states – and the methodology is capable of – deriving ECWL estimates at tide gauge locations, the paper does not present these. This global focus also leads to weak conclusions in the abstract. Larger allowances in the tropics than elsewhere is already known and not new (e.g. Slangen et al. 2018, Vousdoukas et al. 2018). The large increase of ECWL even under modest sea level rise is also already known and a consequence of the shape of extreme value distributions. The value of the paper would increase through an in-depth presentation and discussion of the regional estimates, beyond the four locations that are presented in Fig. 2. The data for each tide gauge station (Fig. S6-S10) should be presented in a Supplementary Table or in other digital form to be of value for local stakeholders.

We have expanded the results and discussion section beyond the summary statistics, and we have added a discussion on how the local changes in ESLs are related to the ESL characteristics and MSL changes. However, we are afraid that we did not clearly emphasize the objective of this paper, which is to quantify the impact of the emission scenario and Antarctic Ice Sheet contribution on the expected ESL changes, given the large uncertainty in especially the latter. We have changed the introduction to make our objective more clear, see also our response to the previous comments.

Please also note that all the underlying data and scripts, as well as all ESL changes for each individual scenario, is available here: <https://thomasfrederikse.stackstorage.com/s/MsHhwz2nZjxUEGs>. All this data will be put in a public repository as soon as the paper is in its final form.

3) On the level of global summaries, I think it would be really valuable to better understand the relation between sea level change and ECWL change on an analytical level. The paper presents the numbers, but leaves the reader alone with guessing relations between slr and ECWL change. This could be exemplified, for example, with a figure showing both global sea level and aggregated ECWL evolution and divergence. Maybe even an analytical expression could be derived for local slr vs ECWL change. Vousdoukas et al. 2018 is a good source of inspiration for presenting results.

Thanks for this suggestion, we have added a discussion on the interplay between the local ESL parameters, the MSL changes, and the resulting changes in future ESL events. We have also added an extra figure to the text (Figure 3), which shows the ESL properties of all tide gauges and how these properties relate to the return periods. These figures help to make the direct comparison between the GPD parameters, MSL changes, and the resulting ESL changes. We have also added a discussion and an extra figure (Figure 6) to show the relation between the allowances and MSL changes.

As far as we are aware of, there is no direct analytical expression to obtain the best-estimate return curve, which is required to derive the allowance, given that both the GPD parameters and the MSL change come with uncertainties, and we feel that deriving an analytical expression of the allowances and amplification factors falls outside the scope of the present manuscript. Such an expression does exist for the Gumbel distribution and an uncertain MSL change (e.g. Hunter, 2012), but given the large number of variables that go into the computation, a numerical Monte-Carlo simulation is probably our best bet.

4) The authors use the generalized pareto distribution to fit extreme water level distributions instead of the Gumbel distribution used in most of the earlier works, e.g. Hunter et al. 2017. The sensitivity of the paper's results to this choice is not discussed. This makes it difficult to compare to earlier works. Please add. The authors also discuss that many places AF100 of 100 or larger, meaning that former 100y events occur sub-annually. How does the methodology deal with sub-annual events? How does it handle mean sea level rising above the 100y ECWL? Does it occur at some places?

We have chosen the GPD over the Gumbel distribution because recent paper from Wahl et al., 2017 and Buchanan et al., 2017 have done comparisons between multiple extreme-value distributions, and conclude that the Gumbel distribution could lead to substantial biases, and because the GPD distribution uses more of the available data, we consider it to be the preferred distribution to model extreme events. The methods section contained a discussion about this choice, and we decided to move it to the main text to ensure this discussion is picked up by the reader.

There are indeed many stations for which the MSL changes will result in amplification factors above 100. In the current methodology, the length of the observational record and the GPD threshold put an upper limit on the period for which we can compute the return level. The combination between a minimum record length of 20 years and a threshold of 99.7% guarantees that we can always compute the return curves for at least annual events, and we can compute 100-year amplification factors of at least 100. On average, the method allows to compute the return height of bimonthly events. For scenarios for which the future return period of the present-day 100-year event is shorter than 1 year, we assign these stations an amplification factor of 'more than 100'. We have added a remark about this in the methods section:

The combination of a minimum record length of 20 years and a threshold parameter of 99.7% guarantees that we can always compute a return curve for events with a return period of 1 year or more. We do not account for sub-annual events, and amplification factors that reach the sub-annual level will result in a 'larger than annual' output.

There are stations for which the MSL change exceeds the present-day 100yr ESL event, especially in the tropics under the high-end scenarios. We have added a figure (Figure 7) and a discussion on the occurrence of this phenomenon. Under the high-end scenario, about 40 % of the stations in the tropics faces such a change, which in practice often means that in the future, mean sea level will be above the highest observed extreme. While the numbers itself do not have a direct implication for stakeholders, we believe that these numbers form a strong narrative for the vast sea-level changes that could occur under higher-end scenarios.

5) I would favor the code underlying this work to be presented open source on one of the known hosting platforms (github ...), with a clear description (readme) of all steps to replicate the analysis, including the sources of additional data downloads.

That is indeed a good idea. The code and readme can now be downloaded from <https://thomasfrederikse.stackstorage.com/s/MsHhwz2nZjxUEGs>, and we will place the code on Github when the manuscript has been accepted for publication.

Minor comments:

L32: The maximum AIS contribution is stated as 75cm. This number is not used in any of the main text figures and text. This needs to be explained.

We have updated the AIS contributions, so this sentence is not in the present version anymore. We computed allowances and amplification factors for a wide range of AIS scenarios. To keep the main text concise, we only show the results within the likely range in each figure, while the results for the full range of scenarios can be found in the supplement and in the data supplement.

L47-49: I do not understand this sentence and the “free-board” without further explanation. Please expand on this concept. Maybe merge from Methods.

We have removed the term ‘freeboard’, which may be confusing, and we have added an explanation on why the allowance exceeds the MSL change:

Both future sea-level changes and the present-day return curves come with an uncertainty. To account for this uncertainty, the future return curve shows the expected return frequency under the uncertain present-day return curve and MSL changes. Because the relationship between the return height and return frequency is not linear, a symmetric uncertainty in the MSL change causes an asymmetric uncertainty in the expected change in the return frequency of a specific height above present-day MSL. This asymmetry results in an expected future return curve that lies above the median future return curve. Therefore, both AF and AI are larger than expected from the median MSL change. For locations where the uncertainty in future MSL changes is large and where small MSL changes result in large frequency amplifications, this asymmetry causes a larger difference between the allowance and the expected MSL change. Therefore, the allowance can be used as a guideline of required changes in the design height of coastal infrastructure under an uncertain change of mean sea level.

L58-59: why are there no confidence intervals shown for Brest and Gan? Reads like a physical explanation, but it is just a fit to a function, right?

Confidence intervals are small for negative shape parameters, and hence, the CI’s are shown but not visible on the scale. We have added a remark about this in the caption of the figure.

Fig. 2: The lines for Brest and Gan are very flat. This means that for a certain (small) increase of mean sea level, a mapping to a (shorter) return period cannot be established. This may be just an effect of the y-scale of the figure or a weakness of the methodology. It should be discussed.

When the return curve is almost flat, a small MSL change indeed causes a very large amplification factor. As discussed below point 4, for the situations where the ‘horizontal arrow’ does not have a left bound, we assign a ‘larger than 100’ value to the 100-year amplification factor.

L74: Relatively? Do not understand. “slight” missing?

This sentence has been removed.

L76: How can I have a factor AF100 of 100 or more? Do you resolve sub-annually?

See also our response to point 4: we do to some extent solve sub-annually, but that depends on the individual record length.

L80: This statement is already clear from the author’s choice of future AIS contribution, which has the same magnitude as the sum of all other components.

L89-L90: Meridional overturning circulation: How does it enter into calculations?

To avoid confusion, we have removed the explicit reference to the overturning circulation, but its contribution enters via the mean and spread of the ocean dynamics term from the AR5 projections, which are based on the CMIP5 ensemble.

L108-109: It would also reduce the risk of a high Antarctic contribution. This is not reflected in the author’s choices of climate-independent Antarctic ice loss.

We have added a remark about this to the conclusions:

Both future emissions and the fate of the AIS are poorly constrained, and as a result, allowances typically vary from a few decimetres to one-and-a-half meter by the end of the century within the likely range of AIS

contributions, while changes that are substantially larger cannot be ruled out. A reduction of greenhouse gas emissions substantially lowers the required allowances, and limits the likely range of the AIS contribution, although the uncertain AIS contribution could still cause allowances on the order of 50-100 cm. However, a reduction will not only result in smaller allowances, but will also increase the available time for coastal areas to adapt to changing ESL return curves, although for some locations, especially many open-ocean islands, large increases in return frequencies will occur within a few decades, regardless of the Antarctic contribution to MSL and the emission scenario.

L129: Do you mean Fig. S4 here?

Fixed

L148: Tide gauge data: how many locations are used finally?

460, number has been added to the methods section.

L163-L164: Reference to figures seem wrong.

Fixed

L186-L204: This discussion can easily be missed. It may be better placed at the end of the main text.

Good idea, We moved this section to the main text.

References:

Buchanan, M. K., Oppenheimer, M., & Kopp, R. E. (2017). Amplification of flood frequencies with local sea level rise and emerging flood regimes. *Environmental Research Letters*, 12(6), 064009.

DeConto, R. M., & Pollard, D. (2016). Contribution of Antarctica to past and future sea-level rise. *Nature*, 531(7596), 591.

Hunter, J. R., Woodworth, P. L., Wahl, T., & Nicholls, R. J. (2017). Using global tide gauge data to validate and improve the representation of extreme sea levels in flood impact studies. *Global and Planetary Change*, 156, 34-45.

Slangen, A., van de Wal, R. S., Reerink, T. J., de Winter, R. C., Hunter, J. R., Woodworth, P. L., & Edwards, T. (2017). The impact of uncertainties in ice sheet dynamics on sea-level allowances at tide gauge locations. *Journal of Marine Science and Engineering*, 5(2), 21.

Vousdoukas, M. I., Mentaschi, L., Voukouvalas, E., Verlaan, M., Jevrejeva, S., Jackson, L. P., & Feyen, L. (2018). Global probabilistic projections of extreme sea levels show intensification of coastal flood hazard. *Nature Communications*, 9(1), 2360.

Reviewer 3

This paper uses data from tide gauges to derive return levels for extreme water level events, then considers how these return levels are likely to change with sea-level rise according to certain RCP projections. Antarctic ice loss is treated separately, as scenarios for varying losses from one location in Antarctica.

Results are presented as the "frequency amplification factor" AF100, ie are saturated when a "1/100" year event becomes an annual event, or as AI100 "sea-level rise allowance" - change of 100yr extreme level. This presentation may be useful in communicating the global risk. However the paper contains little discussion of these results.

I recommend that for publication in Nature Communications the paper be revised to either highlight the mechanisms behind the spatial variation, and hence enable generalisation to data-poor regions, or else to provide a data set or maps more directly useful for coastal planners, perhaps exploring some interesting cases.

We have added discussions, as well as extra figures (3 and 6) to the main text to discuss to which extent our results can be extrapolated to data-sparse regions. We now also discuss in more detail how the specific local ESL affects the relationship between local MSL changes and future ESL changes. See also our response to the remarks by reviewer #1.

The explanations around figure 2 are very good - perhaps the most useful part of the paper. Since the shape of the return curve is dependent on tidal range, which is relatively well known (eg line 63), and on cyclone likelihood, a map of tidal range and cyclone tracks may be useful. Can fig 3 then be predicted from these? Or could you indicate which gauges globally are similar in characteristics to the examples 1/2/3/4? Maybe plot all gauges that are like (1) on one map, and all those like (4) on another? For example, is the block of similar results along Alaska/west Canada a direct result of the high tidal range?

We agree with the reviewer that it would be beneficial if we could group locations into specific regimes. However, since the number of parameters that have a serious impact on the ESL changes is quite high, such a grouping would be difficult to establish and could lead to some arbitrary choices. Since the future changes in ESL events thus depend on this multitude of factors, predicting these changes from the tidal range and cyclone occurrence is therefore also not directly feasible.

To explain this spatial heterogeneity better, we have added extra figures and texts to the manuscripts: Figure 3 and the accompanying text discuss the impact of the specific extreme-value characteristics on its changes under future sea-level change. In this discussion, we also show that for example the high tidal range around Alaska results in a large location parameter, but also a high scale parameter, which is a likely a result of local storminess rather than the large tidal range, and results in comparably small amplification factors.

In combination with a new figure (Figure 6), we also added a discussion on whether the local estimates can be extrapolated to locations for which no tide-gauge data is available.

There is a significant problem with the presentation of the data as a scatter-plot at tide gauges (Fig 3, 4, S5-S12), in that even zooming in you cannot see some data points because they are obscured by their neighbours. For example there are points on the US east coast, and in Europe, just peeking out, which are very different from their neighbours, and it is not clear which they are or whether they are important.

The scatterplot order appears to be alphabetical by gauge, so is pseudo-random with location, but this method of plotting can give a false impression of spatial patterns and is useless for looking up individual locations in crowded areas. Can the authors improve the plot to avoid this overlap? An online version of the plots with smaller dot size compared to the coastline would help, as would use of transparency.

We have updated all plots to improve the visibility. For the scatterplots in particular, we decreased the dot size, removed the bounding line, and added transparency to reduce the obscuration of some stations. Furthermore, the underlying data is also available in the data supplement.

There are indeed some large spatial variations in the ESL properties over short distances. In Figure 3, we now show the uncertainty in the present-day return periods, which shows that outside the areas with a positive shape parameter, these local deviations are often larger than the uncertainty. We made a remark about this in the text:

The uncertainty in the derived return heights (Figure 3f) depends on both the scale and shape parameters and the record length, and is generally on the order of one to three decimeters, except for locations with a positive shape parameter, which results in an uncertainty that is about an order of magnitude larger. The spatial variations visible in Figure 3e are generally larger than the uncertainty, except for regions with positive shape parameters, which implies that these spatial variations are not an artefact of the uncertainty.

The uncertainty of the return curves is very variable between gauges, because of the varying length of data as well as the different weather and tide patterns. This is to some extent captured in figure S6 but this is hard to interpret. Could you plot the uncertainty at the 100yr level, to give some indication in the confidence in these results by location? Are those locations that differ from their neighbours those with high uncertainty, or is there a lot of local spatial variability?

Yes, we do have a confidence interval for the amplification factors and the allowances. However, we have chosen to not show them in the main text, because the methodology is based on a best-estimate return curve, which explicitly takes the uncertainty in both the MSL changes and in the ESL parameter estimation into account. We have added a more complete explanation for this choice into the main text:

Both future sea-level changes and the present-day return curves come with an uncertainty. To account for this uncertainty, the future return curve shows the expected return frequency under the uncertain present-day return curve and MSL changes. Because the relationship between the return height and return frequency is not linear, a symmetric uncertainty in the MSL change causes an asymmetric uncertainty in the expected change in the return frequency of a specific height above present-day MSL. This asymmetry results in an expected future return curve that lies above the median future return curve. Therefore, both AF and AI are larger than expected from the median MSL change. For locations where the uncertainty in future MSL changes is large and where small MSL changes result in large frequency amplifications, this asymmetry causes a larger difference between the allowance and the expected MSL change. Therefore, the allowance can be used as a guideline of required changes in the design height of coastal infrastructure under an uncertain change of mean sea level.

We also added a discussion on the spatial variations in the GPD parameters, and added an extra figure (Figure 3), which explicitly shows these parameters. This figure shows that while there are some regions that show homogeneous GPD parameters, large local deviations do occur along many coastlines. The figure also contains the confidence interval for the return period, which shows that the spatial heterogeneity is generally larger than the uncertainty estimate. We have also added a figure and discussion that shows the effect of these sources of uncertainty on the estimated allowance (Figure 6).

It is not very clear why the ice-loss scenarios are all based on changes to Thwaites glacier. I'm not a cryosphere expert - are these a fair reflection of the likely 21st century scenarios? Would other likely scenarios substantially change the result? What about Greenland? Is Greenland ice loss included in the other RCP maps?

While it is indeed unlikely that all Antarctic mass loss will come from the Thwaites glacier, it is now, combined with some other glaciers in the Amundsen Sea Sector, the largest contributor to sea-level rise, and due to its retrograde slope, it is one of the major candidates for future ice-sheet instabilities (Shepherd et al., 2018), and therefore, we have chosen to assume that all mass loss comes from Thwaites. Since almost all locations are at least a few 1000km away, the exact location of the mass loss in the Amundsen Sea Sector has a limited effect on local MSL changes at the tide-gauge locations (e.g. Mitrovica et al., 2018). We have updated the discussion to the methods section to highlight this:

We assume that all Antarctic mass loss occurs at Thwaites Glacier basin. This basin currently shows the largest mass-loss rate and has the potential of ice-sheet instabilities. Other nearby regions along the Amundsen Sea Sector are vulnerable as well, but since almost all tide-gauge locations considered in this study are in the far field, small changes in the geometry of the mass loss are unlikely to significantly change the results.

The mass loss from the Greenland Ice Sheet is incorporated into the projections. We use the IPCC AR5 projections for Greenland, which also includes an estimate of the associated spatial pattern. This is described in the methods section as well.

Return levels calculated at tide gauges could include many local effects, including tidal range and wetting-and-drying of low-lying land. These are ignored in the calculations, which assume a linear addition of the extra water. This is probably necessary in a global study but should be explicitly stated. Likewise the assumption of no future change in extremes due to tropical cyclones should be made very explicit.

We had a discussion about this in the methods section, but we have added an extra sentence in the main text to make this assumption more explicit.

Another assumption in the statistical method is that of stationarity, i.e., the assumption that the local ESL climatology does not change in time.

Can you avoid the unusual acronym "GRD"? SAL (self-attraction and loading) is more commonly used, although to be fair that doesn't encompass rotation.

This new abbreviation comes from the recent paper from Gregory et al (2019) who aims to define standard definitions for many terms related to sea-level rise, which include GRD effects, as well as Extreme Sea Level (ESL) events. We have tried to adhere to this terminology throughout the paper, and we have added a reference to this paper when the term is introduced.

typo fig 2 "Maldives".

Thanks, fixed!

REVIEWERS' COMMENTS:

Reviewer #1 (Remarks to the Author):

The authors improved the clarity of the manuscript and now base their projections on Antarctic ice loss on estimates recently published. This is appreciated. The abstract was complete rewritten, which is not motivated nor mentioned in the author responses. The new abstract makes a point even stronger that I commented on already, but again here more explicitly: treating scenario uncertainty, i.e. the societal choices on future carbon emissions, and process uncertainty, i.e. our incomplete understanding of Antarctic ice loss, as equivalent.

As I read from the author responses, this is explicitly wanted. I see the risk that this is interpreted wrongly by many readers, i.e. in simple terms: „uncertainty of Antarctic ice loss makes climate change irrelevant for SLR“. The authors should seek to mitigate this risk if the manuscript is revised. I think this point needs to be addressed upfront, not only in the discussion section. I understand that from the perspective of a coastal planner the presented approach is justified. The manuscript however does not address coastal planners, but in its current form rather a readership on global change.

The methodological choices are now very close to the IPCC SROCC report. Can the relation to the report be discussed in the paper? Which parts are the same, where is it different? This is important for readers to know.

On a more general note, I see reviewer's remarks mostly addressed by adding more detail to the manuscript. This makes it a long read now, and I suggest to critically revisit and cut where possible. The authors should keep the broad range of readership in mind. Examples are: allowance is not explained in the abstract. Figure 2 Loc, scl, shp need clearer naming and relation to the caption.

I46: „quite some time“ is imprecise

I66: refer again to Fig. 1

Fig. 1 clearer labeling: XX cm can be confused with total SLR, but is only AIS

I77: correct „comma dot“

I133, „35% and 63% of the global and tropical locations“, respectively

I135: „under RCP2.6 the majority (72%) of the tropical locations will face an AF 100 of 100 or higher, while under RCP8.5, this is the case for about half (52%) of all locations.“

Do not mix tropical and all locations in such way. Difficult to grasp.

I188: wording: „regardless of emission scenario“. Emissions still have a strong influence.

I200: wording: „regardless of the chosen scenario“. Same here, please find better wording. Sounds like emission scenario is not relevant, which is not true.

Reviewer #3 (Remarks to the Author):

Review of Frederikse et al, Antarctic Ice Sheet and emission scenario controls on 21st-century extreme sea-level changes.

This is my 2nd review of this paper. Thank-you for the improvements you have made. I still feel that it rather misses the opportunity to present the mechanisms between the spatial variability with less reliance on readers' prior knowledge of the field. For example there could be a map of tidal range alongside figure 3 panel of location parameter, or hurricane tracks alongside the shape parameter. However it can be published. Here are a few minor suggestions.

Figure 2:

(i) I'm not entirely convinced that tropical/extratropical by latitude is the best division here, though I wouldn't insist on changing it since you have used it so much elsewhere.

Equatorial/tropical/extratropical may be more useful as Saipan and Galveston appear to have much more in common than Saipan and Gan, which is not subject to cyclones. A map panel somewhere - perhaps in figure 3 - of hurricane tracks or maximum wind speed over the last few years, would be interesting to see the correspondence with shape parameters.

(ii) In the Example panel, the red arrow should only be up, and the green only left, to indicate the sense of the amplification factor. (It is not impossible that a site with large VLM could still have a negative amplification).

(iii) In the caption, please add μ , λ , ξ . also parameter \rightarrow parameters.

(iv) In the map panel & caption, use "site" instead of "location" to reduce confusion with "location" μ .

(v) Please set the vertical axes of panels 1-4 to use round numbers - it will be easier to read and less cluttered, and more immediately obvious that the vertical scale differs.

(vi) I don't see any note that the observed extremes at Saipan are a long way from the curve.

(vii) I think "height above present-day MSL" needs to be made explicit on the vertical axes.

Otherwise they could be read as height above MSL *at that date*, and you have changing MSL. Perhaps it would help to have a specific notation for "MSL at 1986-2005". MSL₂₀₀₀? ... come to that you could also use "baseline MSL" instead of "present-day MSL", since it's already 20 years out of date!

Line 71 : suggest: "the location parameter, a height which roughly..."

Line 74: see comment (i) on Figure 2.

line 84: spatially heterogeneous == vary from place to place ? rephrase.

paragraph 105-112 & figure 6. Sorry, this is baffling. I have no idea why AI is not equal to Δ MSL. Rewrite.

Figure 5: The colour scheme is a bit confusing as dark/light is inverted from figure 1.

line 270: account for

line 280: We chose this approach...

line 308: At Saipan λ is very small - is this equation still well defined?

line 312: event occurs only approximately 6 times a year. The number depends on the tidal range, with a large range and few storms it could be 26 times, if there's no tide and just a seasonal cycle it could be once.

line 316: Are you sure the curve has an upper bound, or is it just asymptotic to the linear? If so, what is it?

Reviewer #1

The authors improved the clarity of the manuscript and now base their projections on Antarctic ice loss on estimates recently published. This is appreciated. The abstract was complete rewritten, which is not motivated nor mentioned in the author responses. The new abstract makes a point even stronger that I commented on already, but again here more explicitly: treating scenario uncertainty, i.e. the societal choices on future carbon emissions, and process uncertainty, i.e. our incomplete understanding of Antarctic ice loss, as equivalent.

As I read from the author responses, this is explicitly wanted. I see the risk that this is interpreted wrongly by many readers, i.e. in simple terms: „uncertainty of Antarctic ice loss makes climate change irrelevant for SLR“. The authors should seek to mitigate this risk if the manuscript is revised. I think this point needs to be addressed upfront, not only in the discussion section. I understand that from the perspective of a coastal planner the presented approach is justified. The manuscript however does not address coastal planners, but in its current form rather a readership on global change.

We fully agree that this mis-interpretation has to be avoided. We have re-phrased this sentence in the abstract to explicitly state that emission reductions do matter:

Strong emission reductions lower the risk of large ESL changes but due to AIS uncertainties, cannot fully eliminate the risk that large increases in frequencies of ESL events will occur.

The methodological choices are now very close to the IPCC SROCC report. Can the relation to the report be discussed in the paper? Which parts are the same, where is it different? This is important for readers to know.

The underlying methodology to compute changes in the return curve given a MSL change is equivalent to the method SROCC report. We have added a sentence to the introduction to make this clear:

This method is equivalent to the method used in ref.24.

On a more general note, I see reviewer's remarks mostly addressed by adding more detail to the manuscript. This makes it a long read now, and I suggest to critically revisit and cut where possible. The authors should keep the broad range of readership in mind. Examples are: allowance is not explained in the abstract. Figure 2 Loc, scl, shp need clearer naming and relation to the caption.

We went through the manuscript to address any unclarities. We have removed the sentence about amplification factors from the abstract, and we have added the meaning of the abbreviations to the caption of Figure 2.

l46: „quite some time“ is imprecise

The 'quite some time' refers to the time needed to substantially improve our understanding of the fate of the Antarctic Ice Sheet, to an extent that the uncertainties of its contribution to future sea level become smaller and well-constrained. We don't know how long this will take, but our best guess is that this problem is not going to be resolved in the very near future. Hence 'quite some time'.

l66: refer again to Fig. 1

Added

Fig. 1 clearer labeling: XX cm can be confused with total SLR, but is only AIS

Added AIS to the labelling, and repeated that for all subsequent figures.

l77: correct „comma dot“

Fixed

I133, „35% and 63% of the global and tropical locations“, respectively
Fixed

I135: „under RCP2.6 the majority (72%) of the tropical locations will face an AF 100 of 100 or higher, while under RCP8.5, this is the case for about half (52%) of all locations.“

Do not mix tropical and all locations in such way. Difficult to grasp.

We agree. We have split up this sentence:

Adding a 10cm AIS contribution, which corresponds to the upper end of the likely range for this scenario, results in a further increase in frequency amplification factors: under RCP2.6, almost half (44%) of all locations, and the majority (72%) of the tropical locations will face an AF100 of 100 or higher. Under the RCP8.5 scenario with a 10cm AIS contribution, this is the case for more than half (52%) of all locations, and for more than three-quarters (77%) of the tropical locations.

I188: wording: „regardless of emission scenario“. Emissions still have a strong influence.

We have changed the sentence to avoid this possible mis-interpretation:

This analysis shows that a strong reduction in greenhouse gas emissions reduces the allowances and the amplification factors, but even under the RCP2.6 scenario, some locations, particularly in the tropics will see large amplification factors halfway this century.

I200: wording: „regardless of the chosen scenario“. Same here, please find better wording. Sounds like emission scenario is not relevant, which is not true.

We have removed the first part of the sentence, which now reads:

The allowances required to keep the exceedance risk at present-day levels do depend on the emission scenario and AIS contribution, even halfway through the century.

Review of Frederikse et al, Antarctic Ice Sheet and emission scenario controls on 21st-century extreme sea-level changes.

This is my 2nd review of this paper. Thank-you for the improvements you have made. I still feel that it rather misses the opportunity to present the mechanisms between the spatial variability with less reliance on readers' prior knowledge of the field. For example there could be a map of tidal range alongside figure 3 panel of location parameter, or hurricane tracks alongside the shape parameter. However it can be published. Here are a few minor suggestions.

Figure 2:

(i) I'm not entirely convinced that tropical/extratropical by latitude is the best division here, though I wouldn't insist on changing it since you have used it so much elsewhere. Equatorial/tropical/extratropical may be more useful as Saipan and Galveston appear to have much more in common than Saipan and Gan, which is not subject to cyclones. A map panel somewhere - perhaps in figure 3 - of hurricane tracks or maximum wind speed over the last few years, would be interesting to see the correspondence with shape parameters.

We have discussed this idea in a previous stage during the writing process of this as well. The reason why we thought that adding this extra information would not be beneficial is twofold: the first reason is that determining the causes of local ESL characteristics is a bit outside the scope of this paper, although we fully agree that a study between hurricanes and ESL characteristics on a global scale would be highly beneficial for the sea-level science community. The second reason is that such maps may suggest a simple link between hurricanes and shape parameters, while this link is affected by a multitude of local factors, such as the direction of the shoreline relative to the storm tracks, local bathymetry etc. Therefore, adding these storm tracks would require a thorough discussion of the link between all these factors to avoid the impression that these quantities are one-to-one related, which we'd argue, is not in the scope of this manuscript.

(ii) In the Example panel, the red arrow should only be up, and the green only left, to indicate the sense of the amplification factor. (It is not impossible that a site with large VLM could still have a negative amplification).

Fixed

(iii) In the caption, please add μ , λ , ξ . also parameter->parameters.

Added

(iv) In the map panel & caption, use "site" instead of "location" to reduce confusion with "location" μ .

Fixed

(v) Please set the vertical axes of panels 1-4 to use round numbers - it will be easier to read and less cluttered, and more immediately obvious that the vertical scale differs.

Fixed

(vi) I don't see any note that the observed extremes at Saipan are a long way from the curve.

We added a note about this behaviour for stations with positive shape parameters:

Due to the fact that over the record period only a few tropical cyclones have been observed, the return period of these events is poorly constrained, which results in a poorly-constrained return curve, which is visible from the large confidence intervals.

(vii) I think "height above present-day MSL" needs to be made explicit on the vertical axes. Otherwise they could be read as height above MSL *at that date*, and you have changing MSL. Perhaps it would help to have a specific notation for "MSL at 1986-2005". MSL_2000? ... come to that you could also use "baseline MSL" instead of "present-day MSL", since it's already 20 years out of date!

Also to decrease the clutter, we have removed 'above MSL' from the axis label, and changed 'above' to 'relative to' in the caption.

Line 71 : suggest: "the location parameter, a height which roughly..."

Fixed

Line 74: see comment (i) on Figure 2.

Replaced 'location' with 'site' for all instances throughout the manuscript that do not refer to the location parameter.

line 84: spatially heterogeneous == vary from place to place ? rephrase.

Fixed

paragraph 105-112 & figure 6. Sorry, this is baffling. I have no idea why AI is not equal to Δ MSL. Rewrite.

We agree that is concept is difficult to grasp. We have re-formulated this sentence to clarify why the expected change in the return curve differs from the mean MSL change

Both future MSL changes and the present-day return curve come with an uncertainty. To account for this uncertainty, the future return curve shows the expected frequency ECWL events under the uncertain present-day return curve and MSL changes. A symmetric uncertainty in the MSL change (the vertical axis in Figure 2) causes an uncertainty in the return period for a specific height (the horizontal axis in Figure 2) that is skewed towards shorter return periods. Therefore, the height increase for which the expected return period stays constant is larger than the expected MSL change. This skewness results in an expected future return curve that lies above the median future return curve. Therefore, both AF and AI are larger than expected from the median MSL change.

Figure 5: The colour scheme is a bit confusing as dark/light is inverted from figure 1.

We inverted the color schemes of Figure 5 and 6 to match light=low, dark=high

line 270: account for

Fixed

line 280: We chose this approach...

Fixed

line 308: At Saipan lambda is very small - is this equation still well defined?

We double-checked the values for lambda for all stations, and we didn't find any station with a lambda smaller than 0.025, which is still well above zero, so the equations are still well-defined.

line 312: event occurs only approximately 6 times a year. The number depends on the tidal range, with a large range and few storms it could be 26 times, if there's no tide and just a seasonal cycle it could be once.

Yes, that's right, we have added 'on average' to make this clear.

line 316: Are you sure the curve has an upper bound, or is it just asymptotic to the linear? If so, what is it?

Yes, the GPD has a limit, which can be derived from Equation 3. The probability density becomes zero for arbitrarily large z when the shape parameter is negative. You can derive the value for z for which the probability is zero from Equation 3 by determining the value for z for which $1 + \xi ((z-\mu)/\lambda) = 0$. This gives that for $(z-\mu) > (-\lambda/\xi)$, the probability is zero. Hence, the curve has an upper bound.

Since we do not use the upper bound in the manuscript, we decided not to add this derivation to the methods section.